# Determining the Preferred Orientation of Silver-Plating via X-ray Diffraction Profile

**DOI:** 10.3390/nano11092417

**Published:** 2021-09-17

**Authors:** Taotao Li, Liuwei Zheng, Wanggang Zhang, Pengfei Zhu

**Affiliations:** 1School of Mechanical Engineering, North University of China, Taiyuan 030051, China; 2College of Materials Science and Engineering, Taiyuan University of Technology, Taiyuan 030024, China; zhengliuwei@tyut.edu.cn; 3College of Mechanical and Electrical Engineering, Central South University, Changsha 030051, China; zhupengfei4105@163.com

**Keywords:** preferred orientation, silver plating, Rietveld, quantitative texture analysis, exponential harmonic, crystallite shape

## Abstract

Determining the preferred orientation of plating film is of practical importance. In this work, the Rietveld method and quantitative texture analysis (RM+QTA) are used to analyze the preferred orientation of plating silver film with XRD profile, whose <311> axial texture can be completely described by a set of exponential harmonics index, extracted from a single XRD profile, C_4_^1,1^(0.609), C_6_^1,1^(0.278), C_8_^1,1^(−0.970). The constructed pole figures with the index of the exponential harmonic are following those measured by the multi-axis diffractometer. The method using exponential harmonic index can be extended to characterize the plating by electroplating in a quantitative harmonic description. In addition, a new dimension involving crystallite shape and size is considered in characterizing the preferred orientation.

## 1. Introduction

Practically, polycrystalline plating films reveal some degree of preferred orientation owing to the forces evolving crystal growth [1,2,3]. For this reason, knowledge of the evolution of preferred orientations can provide valuable information to optimize the preparing process [4,5,6].

Generally, preferred orientation is characterized by diffraction-related techniques, such as X-ray diffraction and electron diffraction. For X-ray diffraction, at a fixed θ/2θ, the classical technique to determine preferred orientation is via pole figure analysis, which is rather a complex technique and requires special multiaxis texture attachments, the latter provides sample rotation by changing the tilt angle χ and azimuthal angle φ [7]. Most of the papers in the literature make a rough estimate of visual comparison in the pole figure with the textured specimen and powdered one [8]. 

From thermodynamics, the preferred orientation of plating is governed by the surface (interfacial) and strain energy minimization [9,10]. Usually, most of the crystallites are aligned with the approximate crystallographic orientation, normal to the substrate surface. It is imperative to determine the preferred orientation for revealing the growth behavior. For the prevalent single-wavelength X-ray powder diffractometer, the sample orientations angle Ω, χ, and φ are fixed at (0°, 0°, 90°) or (0°, 90°, 0°) [7]. Thus, extracting the preferred orientation from a single X-diffraction profile based on the Rietveld method and quantitative texture analysis (RM+QTA) seems to be more convenient compared with the traditional pole figure analysis procedure, which usually requires the use of a multiaxis diffractometer. To address the feasibility, the March–Dollase function and exponential harmonics function are employed to correct the preferred orientation of the plating silver film with Rietveld analysis of a single XRD profile, which is further confirmed by the pole figure measurement. The overall aim of this paper is to offer a simpler, quantitative way to determine the preferred orientation of the plating films.

## 2. Experimental and Rietveld Calculations 

### 2.1. Materials Preparation, Electroplating

Silver was coated on copper substrates by an electrochemical method in a typical niacin plating system according to our previous report [6]. Briefly, the copper substrates with of 0.3 mm thickness were polished and ultrasonically cleaned in alcohol first, and then electrochemically coated with Ag films. The copper substrates were taken out from the electrolyte immediately when the color of the silver layer darkened. The electrolyte was a mixture solution (100 mL) containing excess thiourea, CH_3_COONH_4_ (77 g L^−1^), NH_3_·H_2_O (32 g L^−1^), KOH (50 g L^−1^), Niacin (100 g L^−1^), K_2_CO_3_ (77 g L^−1^) and AgNO_3_ (44 g L^−1^). The electroplating time was 40 min under the current density of 0.3 A dm^−2^. All reagents are offered from obtained from Sigma-Aldrich (Sinopharm Chemical Reagent Co., Ltd., Shanghai, China).

### 2.2. X-ray Diffraction Profile and Texture Measurement

Step-scanned profiles (θ/2θ scan mode) were measured with a Smartlab Bragg-Brentano diffractometer (Rigaku Corporation, Akishima-shi, Tokyo, Japan). Data were collected up to 100 ° because Bragg intensities were barely observable beyond this angle. The conventional θ/2θ scan mode was measured to identify the phase and positioning maximum angle. Then, classical pole figures, measured positioning the θ/2θ axis on the maximum of the corresponding reflex, were determined by the scan at 0 of tilting. Then the pole up to 75° was analyzed using a square grid of 5° × 5° × 5° to assure an equal area. The detailed test conditions were summarized in Table 1. 

### 2.3. Rietveld Analysis of X-ray Diffraction Profile

The phase analysis using the diffraction Rietveld program (Materials analysis using diffraction, MAUD 2.91) [11], written and maintained by Luca Lutterotti at the University of Trento [12,13,14,15,16] was used to refine the obtained X-ray diffraction data. The program was initially conducted with no preferred orientation correction. After the refinement had converged, the March–Dollase function and exponential harmonics function were used independently to correct the preferred orientation effect. The least-squares parameter R_wp_ values were used to evaluate the refining results. A value close to 0 indicates a satisfactory fit. If the value is larger, the parameters or the theoretical model may be inappropriate.

#### 2.3.1. Terminology and Definition of Texture and Preferred Orientation

As H.-J. Bunge defined, crystallographic orientation refers to how the crystallites in a volume of crystal (crystal coordinate system) are positioned relative to a fixed reference (specimen coordinate system) [17]. There is a trending pattern in the orientations that are present and a propensity for the occurrence of certain orientations. This crystallographic orientation of the crystallites with the polycrystalline aggregate is known as preferred orientation, more concise texture resulted in the directionally dependent properties.

If all possible orientations of crystallite occur with equal frequency, the orientation dependence will disappear on average. As W. A. Dollase defined, the axially symmetric flat-plate sample can be composed of effective rod- or disk-shaped crystallites. This shaping effect is also named as preferred orientation [18,19,20].

Preferred orientation is a more general terminology in evaluating the above two effects. To the best of our knowledge, there has been no detailed definition of crystallites in plating. In this paper, texture and preferred orientation refer to the more generalized concepts, generalized preferred orientation, namely preferred orientation.

#### 2.3.2. Initial Rietveld Refinements Assuming No Preferred Orientation

The initial Rietveld analysis is based on the crystallographic structure with uniform distribution in the orientation distribution space. In the previous paper, the random orientation of crystallites can be reckoned as uniform distribution. Yet, the shape effect should be taken into consideration because the orientation distribution of needle and plate crystallites are not uniform. For the cubic silver, the randomly distributed silver cube crystallites can be reckoned as having no preferred orientation.

In the following Rietveld analysis, the parameters refined were phase scale factor and the background component of the profiles with eight parameters, the single peak fitted with Caglioti-PV function, anisotropic size-strain parameters, and structural parameters, respectively. With the input of the crystallographic information file, the corresponding unit cell acts as the initial reference frame to compute the diffraction profiles with Equation (1), where s is the scale factor, L is the combined Lorentz and polarization and monochromator factors, P(*o*) is the preferred orientation correction factors and F*_hkl_* is the structure factor for reflection hkl.
(1)$$y_{i}(\theta )\;=\;sLP(o)F_{hkl}^{2}\;+\;y_{b}(\theta )$$

The standard diffraction profile is calculated with no preferred orientation correction, which has a preferred orientation index equal to unity. The preferred orientation samples have higher values, and the preferred orientation index is infinity for single-crystal data. In MAUD, the structure factor F*_hkl_* is extracted with the classical Le Bail method [21].

#### 2.3.3. Rietveld Method with Quantitative Texture Analysis (RM+QTA)

For Rietveld refinement using the preferred orientation correction, the corresponding parameters were refined after the preliminary refinement with no preferred orientation had converged as described above. Then, the March–Dollase function and exponential harmonics function are introduced to correct the preferred orientation. The related functions are introduced below.

In a simple form, diffraction intensities from axially symmetric flat-plate samples, composed of rod or disk-shaped crystallites, can be corrected for preferred orientation with a single-pole figure profile. March–Dollase function Equation (2) is an effective model to correct this kind of preferred orientation [18].
(2)$$P(\alpha )\;=\;{\left(r^{2}\cos^{2}\alpha \;+\;r^{-1}\sin^{2}\alpha \right)}^{-3/2}$$

In Equation (2), *α* is the angle between the preferred orientation direction and the reciprocal-lattice vector direction for the Bragg peak that is corrected. The single parameter r is the adjustable coefficient to reflect the strength of the preferred orientation, which controls the distribution.

In general cases, the preferred orientation can be represented by an expansion in terms of symmetrized harmonics via the orientation distribution function (ODF, *f_s_*(g)), a given function to describe the distribution of crystallite. Among these functions, one way to determining an ODF positivity is to express it with an exponential harmonics function, which results in positive (or zero) values. Such a function can be for instance an exponential with real function arguments h(g). In the Rietveld refinement, the preferred orientation is then completely described by the set of the exponential harmonics coefficients *C_sλ_^mn^*, determined from an iterative process with exponential harmonics description. Then, the complete orientation distribution *f_s_*(g) can be calculated with Equation (4) [22].
(3)$$f_{s}(g)\;=\;e^{h(g)}\;\geq \;0$$
(4)$$f_{s}(g)\;=\;\sum_{\lambda \;=\;0}^{L}\sum_{m\;=\;1}^{M(\lambda )}\sum_{n\;=\;1}^{N(\lambda )}C_{s\lambda }^{mn}T_{\lambda }^{mn}(g)$$

In Equations (3) and (4), *T**_λ_**^mn^* is the series of generalized spherical harmonics and g represents the crystallite orientation.

For the single-wavelength X-ray powder diffractometer used in the present study, the sample orientations angle Ω, χ, and φ are fixed at (0°, 0°, 90°) [7]. The magnitude of the preferred orientation can be evaluated from the preferred orientation coefficient *C_sλ_^mn^*. The preferred orientation can be completed as described by the set of exponential harmonics coefficient *C_sλ_^mn^* determined from the Rietveld refinement exponential harmonics correction.

## 3. Results

The appraisal of the silver film on the copper substrate was carried out with a typical θ/2θ scan for revealing the phase. As shown in Figure 1a, two phases can be indexed as face-centered copper substrate and silver film with space group (Fm3m 225). For the present diffraction peaks of a copper substrate, the information of the silver phase pertains to the entire plating.

In the measured XRD profile, the integrated intensity of {311} is much larger than that in the standard diffraction profiles (Appendix A), which are computed by the X-ray kinematic theory assuming no preferred orientation. The deviation of the intensity ratios from theoretical diffraction profiles is probably due to the <311> preferred orientation, the typical phenomenon of lacking statistics in terms of the number of irradiated crystallites in the plating silver film.

In principle, reasonable refinements of crystal structure and microstructure parameters may be obtained before applying corrections to the diffraction intensities. After refining the scale factor and background, lattice constants (a = b = c = 4.08780(9) Å) of silver films were obtained by refining the basic phase parameters. In the following, the parameters of the quantitative microstructure analysis with Popa rules are listed in the supporting information [23]. Subsequently, the March mode and the description of the exponential harmonic were used independently to extract the preferred orientation index from the Rietveld refinement, regarding the reliability of the formulations and their use in correcting intensities for preferred orientation bias.

Preferred orientation correction can be approximated by a model with fiber symmetry concerning the single XRD profile. Appendix A depicts the calculated XRD profiles derived from Rietveld refinement with the March–Dollase function (Equation (2)). It indicates that the March–Dollase function is not suited to correct the preferred orientation effect in the silver plating. The agreement between the measured and calculated XRD profiles derived from the exponential harmonic’s description (Figure 1b) is superior to that of the March model (Appendix A). The crystallographic R_wp_ factor was the substantially lowest and the detailed Rietveld information of the silver film is summarized in Appendix A. The preferred orientation is then completely described with a set of exponential harmonic indexes, C_4_^1,1^ (0.609), C_6_^1,1^ (0.278), and C_8_^1,1^ (−0.970), with the Rietveld method and quantitative texture analysis. In all, C_4_^1,1^ (0.609), C_6_^1,1^ (0.278), and C_8_^1,1^ (−0.970) can be used to reflect the general <311> preferred orientation in the silver film.

Exponential harmonics are a set of functions, used to represent the orientation distribution of the crystallites and unit cells. They are a higher-dimensional analogy of the Fourier series, which forms a complete basis for evaluating preferred orientation. The exponential harmonic indexes are completely responsible for representing preferred orientation. The exponential harmonic indexes C_sλ_^mn^ can be used to construct and simulate the orientation distribution function with 3D and 2D {111}, {200}, {220} and {311} pole figure as shown in Figure 2a,b Equations (3) and (4). As shown in Figure 2a, there is one sharp ring oriented between 20 to 30° in α with <111> direction. This is consistent with the case in the {200} and {220} pole figure, showing the uniaxial texture in the silver plating. Yet, there are two texture components in {311} pole figure as illustrated in Figure 2a–c, a sharp peak in the center and a broad ring in the range from 30 to 50° in α. The central peak in the {311} pole figure shows the typical <311> texture and the other rings in the range of 30 to 50° also belong to <131> texture.

Then, the classical pole figure measurement is employed to confirm the results refined with Rietveld analysis. By positioning the detector on the center of diffraction peaks 2θ 38, 44.4, 64.5, 77.5°, the measured {111}, {200}, {220} and {311} incomplete pole figures are obtained. With the help of the software TexTools, the orientation distribution function (3D) is constructed as shown in Appendix A. Then, four complete pole figures calculated from ODF analysis are illustrated in Figure 2c. It also can be confirmed that silver film shows uniaxial <311> texture.

It is vital to plot the two-pole axis distributions for comparison with the classical pole figure measurement and those derived by the Rietveld method and quantitative texture analysis (RM+QTA). Figure 3 shows the typical orientation probability distribution vs. α of the silver film. In both the pole-axis distribution, there are three maximums at about 0°, 50° and 85° and two minimums at about 28° and 70°. The intensity of pole-axis distribution of the Rietveld method and quantitative texture analysis (RM+QTA) at 0° is higher than that with classical pole figure measurement, which may be due to the asymmetric distribution of <311> preferred crystallites. There are matching peaks off-axis is pretty much required by the angles between 113 planes. It can be confirmed that the pole figure analysis constructed by the classical pole figure measurement is consistent with that in the Rietveld method and quantitative texture analysis (RM+QTA) to some degree.

It should be noted that the preferred orientation of the silver plating is <311>, which is quite different from the reported texture, such as <111> and <200> preferred orientation. As Thompson proposed, both surface and strain energy minimization control crystallite growth in thin films [24]. In general, thin films of a fcc structure such as Ag show the <111> preferred orientation because the {111} planes have the lowest surface energy, γ_{111}_ < γ_{200}_ < γ_{220}_. Yet, the effect of the strain energy minimization resulted in a <200> texture [10]. The formation of the <311> preferred orientation is a result of the combination of the surface and strain energy minimization.

In all, the refined exponential harmonic indexes are effective to characterize the <311> preferred orientation, which is further confirmed by classical pole figure analysis. With the help of the Rietveld method and quantitative texture analysis, the exponential harmonic indexes are the quantitative ways to determine the preferred orientation extracted from one diffraction profile, which can be extended to uniaxial electroplating films in the prevalent Bragg–Brentano diffraction geometry. 

With the plating times set to 10 min (Appendix A) and 20 min (Appendix A), the preferred orientation can be fully characterized by the exponential harmonic index as shown in Appendix A, which can be used for direct comparison. The increasing trends of the on-axis intensity in (311) pole figure (Appendix A) is consistent with the exponential harmonic index C4^1,1^.

## 4. Discussion

In polycrystalline material, crystallites of different shapes, sizes, and orientations are generally present. The three orientation distribution functions (ODF) and pole figures indeed characterize the polycrystalline state quite well, but still do not completely suffice for all purposes. They do not, for example, consider the crystallite shape. Yet, in the framework of the Rietveld method (RM), quantitative microstructure analysis (QMA) quantitative texture analysis (QTA) can be used to characterize the crystallite shape in an extended manner beyond orientation distribution functions (ODF) and pole figure analysis.

As explained by H.-J. Bunge, texture is directionally dependent. Yet, the preferred orientation, the crystallite shape defined by W. A. Dollase can be directionally independent. Both of them distort the intensity ratios of reflection in the diffraction profiles for the lack of statistics in terms of the number of irradiated crystallites and unit cells. To improve the conditions, it applies to extracting the preferred orientation from the whole diffraction profiles based on the relative intensities with the Rietveld method and quantitative texture analysis (RM+QTA). This is the basis of characterizing the texture of plating in a quantitative description. Thus, it can be inferred that the simulated crystallite in Appendix A with Popa rules shows the component of preferred orientation. The refinement of both quantitative texture analysis (QTA) and quantitative microstructure analysis (QMA) is needed to describe the average crystallite shapes.

The application of the Rietveld method and quantitative texture analysis (RM+QTA) can also be extended to axially symmetric volume-conserving compression and expansion, which show contours of the equal density of poles to some specific planes, a uniform likelihood of finding the (hkl) pole at any azimuth around the pole to (HKL). With the Rietveld method refining preferred orientation, the sample has cylindrical symmetry along the axial direction in the compression and expansion or potential direction in the plating. It is also a major advantage to characterizing preferred orientation with a set of the exponential harmonic index in a quantitative manner. A new dimension of crystallite shape and size is also added to the preferred orientation analysis.

With the traditional pole figure measurement, complete orientation distribution requires several pole figures (typically 1300 measured points per pole figure being necessary for low resolution using a 5° grid) [22], which take nearly two hours. Yet, it is a better way to reduce the time to determine the preferred orientation with the Rietveld method and quantitative texture analysis (RM+QTA).

## 5. Conclusions

In this study, the description of the exponential harmonic, extracted from a whole diffraction profile via the Rietveld method and quantitative texture analysis, provides reliable results for silver plating. The <311> uniaxial texture can be completely described by a set of exponential harmonic indexes, C_4_^1,1^ (0.609), C_6_^1,1^ (0.278), and C_8_^1,1^ (−0.970). Then, the constructed pole figures are confirmed by the classical texture measurement. Therefore, the exponential harmonic approach can be used to extract the preferred orientation information from whole diffraction profiles, which can be extended to characterizing other plating and volume-conserving compression and expansion samples in a quantitative description. In addition, a new dimension of crystallite shape and size is considered in the preferred orientation analysis.

## Figures and Tables

**Figure 1 nanomaterials-11-02417-f001:**
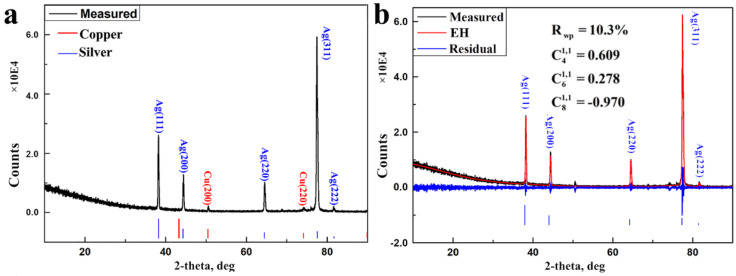
(**a**) Measured θ/2θ diffraction profile of silver plating on the copper substrate; (**b**) the agreement between measured and calculated XRD profile for silver phase following Rietveld refinement with exponential harmonics description (EH) for preferred orientation correction.

**Figure 2 nanomaterials-11-02417-f002:**
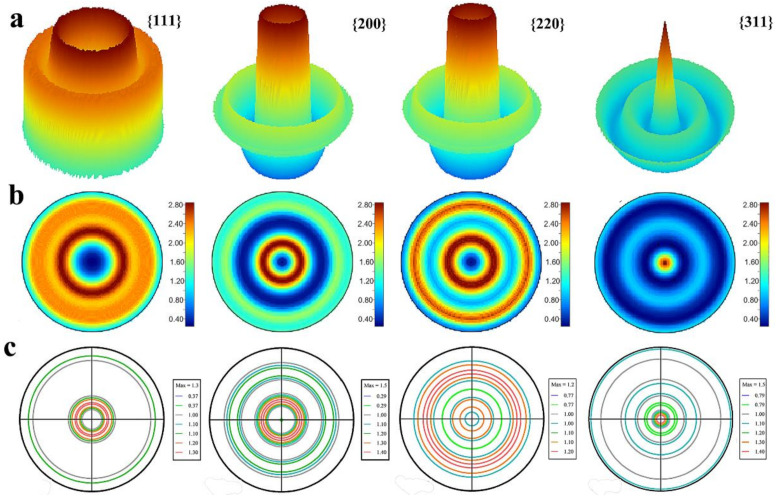
(**a**) The simulated 3D pole figure with the exponential harmonics method in MAUD. (**b**) The simulated 2D pole figure with the exponential harmonics method in MAUD. (**c**) The constructed completed pole figure with the traditional incomplete pole figure measurements with TexTools.

**Figure 3 nanomaterials-11-02417-f003:**
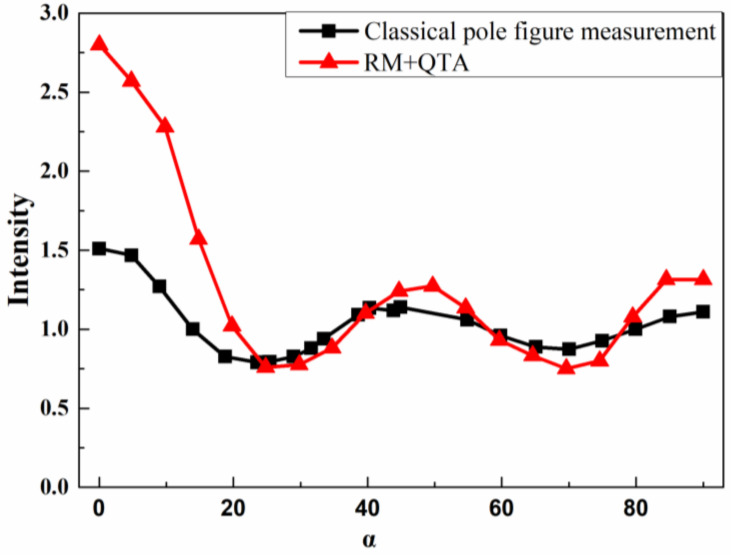
Pole-axis distribution plots for (311) silver film derived from the classical pole figure measurement and Rietveld method and quantitative texture analysis (RM+QTA) (β = 0).

**Table 1 nanomaterials-11-02417-t001:** XRD profiles measurement conditions.

Instrument	Smartlab
**Radiation**	Cu anode tube operated at 40 kV and 30 mA Wavelength: Cu Kα = 1.5418 Å, Cu Kα_1_ = 1.54060 Å, Cu Kα_2_ = 1.54441 Å
**Optics**	Parallel beam (PB)
**Specimen**	Flatted sample
**Detection**	Name: DteX250 (H); Pixel sizes: 0.075
**Scan speed**	20°/min
**Sollerslit**	2.5 deg

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
