# Peer review of "Determining the Preferred Orientation of Silver-Plating via X-ray Diffraction Profile"

_nanomaterials, 2021, doi:10.3390/nano11092417_

Round 1

Reviewer 1 Report

Dear Editor,

the manuscript entitled "Determining the preferred orientation of silver-plating via X-ray diffraction profile" by Li Taotao and co-workers reports on a novel approach to evaluate the anisotropic orientation of polycrystalline films via X-ray diffraction analysis.

The paper is well structured and the results are clearly presented and discussed. The topic reported by the authors might be helpful for a broad audience, thus making it suitable for publication on Nanomaterials.

My only comment concerns the large number of typos that can be found throughout the manuscript. I would therefore suggest the authors to carefully re-read the manuscript and check for typos and grammar mistakes.

Best regards.

Author Response

The response to the reviewers was upload as the attachment. Hope to meet your needs.

Reviewer 2 Report

First, I'll make specific points, then  go to more general issues.

Intro: It's stated that thermodynamically, one expects preferred
orientation along close-packed planes.  This is commonly observed for
fcc metals on amorphous or polycrystalline substrates and would imply
a 111 texture.  However, the observed texture is 311, which is not at
all close-packed.  This curious fact is not discussed.  Is there a
strain-energy argument, say that in-plane strain would nudge the texture
to one in which the relevant elastic modulus is a minimum?

The technical importance of the texture of a silver-plate film is nowhere
mentioned.  Why do you care what the texture is?

Page 3:  What is the thickness of the Ag film?  This is relevant because
the 1/e absorption depth for the CuKa radiation used in pure Ag is 4.5um.
Given that the beam has to come in and out and does so at other than
normal incidence, the observation of any Cu reflections means either
that the film is <~1um thick (at a guess) or has holes in it.  In any
case, there is no discussion of whether the texture results pertain to
the entire film or just a near-surface region.

Equations:  All the equations are shoved over to the left side of
the page, beyond the left margin.  In eq's 2-5, alpha, r and g are
not defined.  This could be fixed by citing a paper such as

Zolotoyabko, Emil. "Determination of the degree of preferred orientation
within the March–Dollase approach." Journal of applied Crystallography 42,
no. 3 (2009): 513-518.

Eq. 4:  Should't the LHS be h(g) instead of f_s(g)?  Also, orientations
are usually denoted with an uppercase omega, not lowercase g.

Bottom of page 4:The equations given do not explain at all the
exponential-harmonics model.  The notation given seems to have come from
a chapter in the International Tables International Tables for
Crystallography (2019). Vol. H. ch. 5.3, pp. 555-580
https://doi.org/10.1107/97809553602060000968 .  The actual method is
much more complicated than explained in the manuscript.  Further, the
variables are not defined.  Only by Googling was I able to find out that
T is a generalized spherical harmonic and that  Eqs. 3-5 are part of an
iterative scheme.  The impression this leaves is that the authors don't
understand the underlying method and simply use MAUD.

Page 5:  This sentence, "It is the typical phenomenon of lacking statics in
terms of the number of irradiated crystallites, the preferred orientation
effect, in the plating silver film. The enhancement of the {311} peak in
the Bragg-Brentano diffraction geometry may be due to the enforcement of
{311} plane." is really hard to understand.  I *think* it
means that the deviation of the intensity ratios from theoretical is
probably due to the 311 texture.

Page 6: The discussion of the Rietveld refinement is incomplete at best.
The values of numbered parameters are given without any discussion of the
physical meaning of these parameters.  What does a negative aniso cryst
size 5 even mean?  What is the shape of the coherent diffracting
domain (don't call it a grain!)?  Is that what's presented in the bottom
of Fig. 1?  If so, it should have a scalebar and it should be stated
which crystal axes the 'prongs' point along.  What is the instrument
broadening compared with the peak widths, and how was that determined?
What is the RMS strain, and in which direction?  Little information on
the physical meaning of these parameters is provided.

Fig. 2:  The reader is invited to compare parts b and c but can't
because they're plotted differently.  In one case, it's a continuous
color map, in the other it's a series of discrete contours.  Surely it
should be possible to export the data from the two programs and plot them
in common format.  By zooming way in, it was possible for me to see that
the 8 z-scales all had common upper and lower limits.  Were they all
replaced with one z-scalebar on the side of the figure, that scalebar
could be made much bigger so the numbers could be more-easily read.

In Figure 2, we can see that there's qualitative agreement between the
measured (incomplete) pole figures and those derived by the new method.
What we can't tell is how quantitative that agreement is.  What are the
FWHMs of the 311 distributions derived from direct pole figure and
the Bragg-Brentano method?

Given the azimuthal symmetry, one could ask a question like the
following:  Consider the population of grains whose 311 axes lie in
the xz plane (z=normal).  Are their -466 axes (ppd 311) distributed
randomly in the plane perpendicular to their 311's, or is there a bias
toward or away from the x-axis?  No question like this was articulated,
yet it's the only reason for working with the complexities of
generalized spherical harmonics.

Page 7:  Is an 'incomplete pole figure' what you get when you do a
theta scan with 2theta fixed?

On the whole, the advantage of using the fancy method on an azimuthally
symmetric system was not demonstrated.

If I'm understanding this paper correctly, the claim is that with just
a single Bragg-Brentano scan, the texture can be fully characterized.
There are 5 peaks showing in the BB scan, and the texture is relevant
only to their intensities.  Further, there's an unknown normalization
factor, so we have effectively 4 data points.  Thus, it makes sense that
only 3 harmonics are derived.  It happens to be the case that the texture
is described, at least qualitatively, by three harmonics, but is that
always the case?  With only one instance, that can't be determined.

Another issue I have is that only one film was examined.  If a given
method of texture measurement is to be useful, it would be in the
comparison of many films made under differing conditions.

There are a number of spelling errors throughout, such as 'Riveted' for
'Rietveld' (AutoCorrect?).

Author Response

The response to the reviewer has been uploaded as attachment. Hope to meet your needs. 

Reviewer 3 Report

Please, see the attached file

Author Response

The response to the viewer has been upload as attachment. Hope to meet your needs.

Round 2

Reviewer 2 Report

The authors have clearly put in a lot of effort in answering my objections, which is much appreciated.  There are still a few points to be cleared up.

I had asked "(1) The technical importance of the texture of a silver-plate film is nowhere mentioned. Why do you care what the texture is?".  This question has not been answered.  Do the wear properties of the film depend on texture?  Literature could be cited.

New autocorrect-induced bug:  In the sentence "The deviation of the intensity ratios from theoretical diffraction profiles is probably due to the <311> preferred orientation, the typical phenomenon of lacking statics in terms of the number of  irradiated crystallites in the plating silver film.", the word "statics" should be "statistics".

Reference 24 (Popa+Balzar) has the wrong year.  The full ref is Popa, N. C., and Davor Balzar. "An analytical approximation for a size-broadened profile given by the lognormal and gamma distributions." Journal of Applied Crystallography 35, no. 3 (2002): 338-346.

The other Popa refs cited in the response letter are not cited anywhere in text, but are needed for the reader to make sense of the parameters.

Presumably the average strain comes out of the Rietveld analysis and could be converted into stress.  As it is, the reader can't see if the stress is compressive or tensile in the plane, which is definitely relevant for practical use of the films.  High tensile stress could cause cracking, high compressive stress could perhaps cause hillock formation, and high stress either way could cause the substrate to bend.

I find the agreement between the classical pole figure and RM+QTA shown in Figure 3 to be less than impressive.  The enhancement of the on-axis distribution of 113 directions over isotropic differs by a factor of 2 between the two methods.  That there are matching peaks off-axis is pretty much required by the angles between 113 planes (35deg, 50.5deg, 63deg, 95deg).

I wonder if the odd shape of the crystallite shown in Fig. S3 is partly an artifact of the way symmetry is treated.  I suspect that the actual shape of a typical crystallite is a needle pointed in the 113 direction that's normal to the surface, but the software assumes the full cubic symmetry of the crystal, thus superimposing copies of the needle pointing in all the 113-family directions.  Thin films often have a columnar structure, so needle-shaped crystals would make sense.  Verifying this could in principle be done by cutting, polishing and etching a cross-section sample and examining it under an SEM, though this wouldn't be required for publication.

Figure S4 seems to show a stack of identical slices, which makes sense for an azimuthally-symmetric distribution.  It would be clearer to show one slice, face-on, as a contour map, with a z-scalebar. 

Table S2 shows that the texture changes with deposition time, but it's not obvious from looking at the numbers what that change looks like.  I suggest replacing the table with a graph like Fig. 3, but with the distributions overplotted for all films.

I suggest that a good procedure for fiber-textured samples might be to take the Bragg-Brentano scan to determine which axis is preferred, then do a single omega scan to get its distribution.

Author Response

Hope to meet your needs.
